# Efficient Discovery of Partial Periodic Patterns in Large Temporal Databases

Rage Uday Kiran [1,2,3,*], Pamalla Veena [4], Penugonda Ravikumar [1,5], Chennupati Saideep [6], Koji Zettsu [2], Haichuan Shang [3], Masashi Toyoda [3], Masaru Kitsuregawa [3] and P. Krishna Reddy [6]

[1] Department of Computer and Information Systems, The University of AIZU, Aizu-Wakamatsu, Fukushima 965-8580, Japan; raviua138@gmail.com
[2] National Institute of Information and Communications Technology (NICT), Tokyo 239-0847, Japan; zettsu@nict.go.jp
[3] Kitsuregawa Laboratory, The University of Tokyo, Tokyo 153-8505, Japan; shang@tkl.iis.u-tokyo.ac.jp (H.S.); toyoda@tkl.iis.u-tokyo.ac.jp (M.T.); kitsure@tkl.iis.u-tokyo.ac.jp (M.K.)
[4] Sri Balaji PG College, Ananthapur 515001, India; rage.vinny@gmail.com
[5] International Institute of Information Technology, Idupulapaya 516330, India
[6] International Institute of Information Technology, Hyderabad 500032, India; saideep.chennupati@research.iiit.ac.in (C.S.); pkreddy@iiit.ac.in (P.K.R.)
* Correspondence: udayrage@u-aizu.ac.jp; Tel.: +81-242-37-2500

**Abstract:** Periodic pattern mining is an emerging technique for knowledge discovery. Most previous approaches have aimed to find only those patterns that exhibit full (or perfect) periodic behavior in databases. Consequently, the existing approaches miss interesting patterns that exhibit partial periodic behavior in a database. With this motivation, this paper proposes a novel model for finding partial periodic patterns that may exist in temporal databases. An efficient pattern-growth algorithm, called Partial Periodic Pattern-growth (3P-growth), is also presented, which can effectively find all desired patterns within a database. Substantial experiments on both real-world and synthetic databases showed that our algorithm is not only efficient in terms of memory and runtime, but is also highly scalable. Finally, the effectiveness of our patterns is demonstrated using two case studies. In the first case study, our model was employed to identify the highly polluted areas in Japan. In the second case study, our model was employed to identify the road segments on which people regularly face traffic congestion.

**Keywords:** data mining; knowledge discovery in databases; pattern mining; periodic patterns

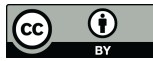

## 1. Introduction

Frequent pattern mining is a valuable and interesting technique for knowledge discovery within data mining. It plays a crucial role in a wide range of real-world applications, such as market basket analysis [1], air pollution analysis [2], traffic congestion analysis [3], privacy-preserving analysis [4], and fraud detection analysis [5]. The primary objective of frequent pattern mining is the identification of frequently co-occurring patterns (items, events, sensor identifiers, etc.) within a group of transactions (i.e., database) based on a user-specified minimum support threshold. For example, when we analyze transactions in a supermarket database, we can see that most consumers who purchased jaggery will also purchase milk as part of an Ayurvedic home remedy treatment for certain types of skin diseases. Another example is an air pollution database; after analyzing transactions, we can discover the frequently polluted geographical areas in which harmful air pollutants (i.e., PM 2.5) are present so as to provide valuable suggestions for people to avoid those areas. In the vast literature, multiple approaches for the discovery of frequent patterns in transactional databases have been discussed. Therefore, mining frequently occurring patterns in transactional databases is valuable and highly important, with several real-world applications. However, one significant shortcoming of these approaches is that the

temporal behavior of the patterns have not been considered and the time of occurrence of transactions in the database has been completely ignored.

Against this background, it is crucial to preserve information about the time of occurrence of transactions in temporal databases. All temporal databases maintain fundamental characteristics, such as: (*i*) the raw data that are available in a transactional database are sorted according to their time of occurrence in ascending order; (*ii*) the time of occurrence of each transaction is not uniform, i.e., the inter-arrival time between any two transactions is not constant; (*iii*) there is a possibility that many of the transactions may occur at the same time. Therefore, we can conclude a notable difference between temporal and transactional databases from these characteristics.

Periodic pattern mining considers the temporal behavior of the items as an important factor in the discovery of interesting patterns in temporal databases. Periodic patterns are classified into full periodic patterns and partial periodic patterns. In the case of full periodic patterns, cyclic behavior is strictly monitored inside the databases and uninteresting patterns are discarded based on user-specific constraint measures, such as maximum periodicity. Based on this measure, when the maximum inter-arrival time (or period) of any pattern is greater than the user-specified value, that pattern is discarded from the full periodic patterns. In the real world, one period is not enough to satisfy the user-specified threshold for declaring an interesting pattern as non-periodic. In the case of partial periodic patterns, some events (or patterns) only occur during a particular point in time (such as during weekends), at a particular time of the day or on a particular day of a month. However, these events do occur regularly. For example, when we consider the supermarket database, most customers purchase meat items frequently and other standard items only during the weekends. When we look at a traffic congestion database, traffic congestion is very high during peak hours of the day, such as from 9 a.m.–10 a.m. or 5 p.m.–8 p.m., at particular locations. Similarly, traffic congestion is very high during the weekends, whereas it may be a bit lower on other days. Even though the full periodic behavior of the pattern is significant to the user, partial periodic behavior also exists in most real-world scenarios. As a result, it is often helpful to mine the partial periodic patterns in temporal databases as well.

In a temporal database, partial periodic pattern mining [6,7] entails the discovery of all patterns that show partial periodic behavior. The discovery of partial periodic patterns in a temporal database comprises two essential sub-tasks: (*i*) assessing the periodic interestingness of a pattern and (*ii*) discovering all of the partial periodic patterns in the given temporal database. While the second sub-task can be solved using a variety of pattern-growth algorithms [8,9], the first sub-task is non-trivial and challenging due to the following reasons:

1. The temporal occurrence information of an item in a database has been completely overlooked in most earlier studies on periodic pattern mining [10–18]. These studies implicitly assume that transactions in a transactional database occur uniformly. Researchers need a model that considers the actual temporal occurrence data of an item in order to discover partial periodic patterns in temporal databases;
2. In a temporal database, it is possible that the time of occurrence of any two consecutive transactions is not uniform and that some of the transactions may share a common time stamp. Therefore, a the regularly occurring behavior of a pattern must be determined with both its support and inter-arrival times in the database. Unfortunately, existing periodic pattern models have only considered the support of a pattern when determining its interestingness [19]. As a result, we need to look into new metrics that can determine the interestingness of a pattern by considering both its support and temporal occurrence data.

Furthermore, the combination of the two tasks mentioned above has significant consequences. It has to be noted that the transformation of a temporal database into a transactional database through the merging of transactions with a common time stamp must be avoided because it causes the subsequent issues:

- **Type I issue:** merging transactions with the same time stamp can result in the loss of the actual *support* of a pattern. This may result in an interesting pattern not being identified as a partial periodic pattern.

  **Example 1.** *Let us consider any two transactions in a market basket database, e.g., {Bread, Jam} and {Bread, Milk}, which occurred at the same time stamp, e.g., 9:00 h. When we merge these two transactions into a single transaction, i.e., {Bread, Jam, Milk}, then we lose the actual support of Bread in the accurate data. This may result in the partial periodic patterns that involve the Bread item being missed.*

- **Type II issue:** merging transactions with a common time stamp creates false correlations (or associations) between the items, which may result in an uninteresting pattern being identified as a partial periodic pattern.

  **Example 2.** *Continuing with the previous example, merging the transactions with a common time stamp induces an incorrect correlation between the Jam and Milk items. This may then identify the uninteresting pattern of {Jam, Milk} as a partial periodic pattern.*

The model for finding partial periodic patterns in a temporal database was first described in [6]. In this paper, we present the accuracy of our 3P-growth algorithm. This paper extends the related work by extensively reviewing the current literature. More importantly, the experimental results section (Section 5) was significantly expanded by considering additional databases. This paper shows that the 3P-growth algorithm is efficient in terms of memory and runtime and is highly scalable in all databases, irrespective of the minimum period-support (minPS) and periodicity (per). Finally, we show the usefulness of our model with the help of two case studies: the first regarding air pollution analytics and the second regarding traffic congestion analytics.

The major contributions of this paper are as follows:

- We introduce a new model for finding partial periodic patterns in temporal databases;
- A new measure called period suppor is proposed to determine the periodic interestingness of a pattern in a database. In contrast to existing support-based measures, the period support measure takes the number of cyclic repetitions into account when determining the interestingness of a pattern in temporal databases. When the inter-arrival time is less than the user-specified period, it is considered cyclic (or periodic);
- We propose a 3P-tree that consists of two important components: a linear list-based data structure, named 3P-list, and a non-linear tree-based data structure, named a prefix tree. We also present the Partial Periodic Pattern-growth (3P-growth) algorithm, which is a pattern-growth algorithm that can be used to find entire sets of partial periodic patterns in temporal databases;
- The results from substantial experiments on both real-world and synthetic databases show that the 3P-growth algorithm is memory- and runtime-efficient and highly scalable;
- Finally, we demonstrate the usefulness of our model using two case studies: one on air pollution analytics and one on traffic congestion analytics.

The remainder of the paper is organized as follows. Section 2 discusses the related work. The proposed model for finding partial periodic patterns is described in Section 3. Section 4 introduces our algorithm for discovering all of the partial periodic patterns in a database. Section 5 discusses the findings of the experiments. Finally, Section 6 concludes the paper by outlining future research directions.

## 2. Review of Literature

In this section, we discuss the existing literature on finding frequent patterns. We then discuss the existing literature on finding periodically occurring patterns.

### 2.1. Frequent Pattern Mining

The discovery of frequent patterns has numerous real-world applications [1–3,5] in web and data mining technologies. Several approaches [20,21] for discovering valuable and interesting patterns in large databases have been discussed in recent decades. A recent survey on frequent pattern mining can be found at [22]. Most of the existing approaches utilize measures that are related to the support (or frequency of occurrence) as the primary criteria for discovering interesting patterns in transactional databases. However, each measure contains a selection bias that exaggerates the importance of a valuable and the interesting pattern. Therefore, no universally accepted optimum measure exists for assessing the knowledge of patterns in any database. Instead, researchers have proposed criteria for choosing measures that depend on the needs of the user and/or the application [19]. Unfortunately, all of the earlier approaches entirely ignored the temporal occurrence information of patterns in databases. For example, when considering an air pollution database, it is crucial to identify the sets of frequently polluted areas (patterns) and the time of occurrence of the pollution in those sets. Hence, researchers have considered the time of occurrence (periodicity) of a pattern to be one of the primary measures for discovering the periodic patterns in temporal databases.

### 2.2. Periodic Frequent Pattern Mining

Ozden et al. [23] designed two novel algorithms, named the sequential and interleaved algorithms, to discover the temporal behavior of patterns in transactional databases. The authors used cycle pruning, cycle elimination, and cycle skipping techniques to effectively discover the full cyclic behavior of patterns by designing cyclic association rules. For this, the complete dataset was divided into disjoint sub-sets based on the time stamp information of each transaction so as to complete the mining process. The non-cyclic patterns could then be pruned from these sub-sets with the help of the proposed pruning techniques and the authors claimed that they could complete the mining process as quickly as possible.

Tanbeer et al. [17] designed a novel periodic frequent pattern-growth algorithm to discover the full periodic frequent patterns in transactional databases. The authors also introduced a novel tree-based data structure, named periodic frequent pattern tree (PF-tree), to store patterns and complete the mining process. The PF-tree had a particular node called a tail node, which was used to maintain a list of the transaction identifiers of the patterns. While pruning these nodes, this list was moved to its parent node to preserve the occurrence information. The authors claimed that the complete mining process was efficient. They generated full cyclic periodic frequent patterns using a novel maximum periodicity measure and a support-based measure in transactional databases.

Amphawan et al. [18] designed a non-support metric-based algorithm, named mining top-k periodic frequent patterns (MTKPP). The authors used an efficient list-based data structure, named the top-k list structure, to maintain the k periodic frequent patterns by only scanning the transactional database once. The MTKPP algorithm used an efficient best-first strategy to discover the top-k periodic frequent patterns in these top-k lists.

In the past, we have also made several attempts to discover full periodic frequent patterns in transactional databases [24–29]. Uday et al. [24] designed a model-based pattern-growth approach to discover rare full periodic frequent patterns in non-uniform transactional databases. The authors used an efficient list-based tree data structure, named the multi-constraint periodic frequent pattern tree (MCPF-tree), to effectively complete the mining process. The MCPF-tree had an MCPF-list and a prefix tree to preserve the transactional identifiers of the patterns. The authors also used two novel constraint measures, named minimum item support and maximum item periodicity, to overcome the combinatorial explosion problems that occur while performing the mining. Furthermore, a novel method was proposed that could dynamically assign the maximum item periodicity value of any pattern. Finally, the authors claimed that the designed approach could extract rare full periodic frequent patterns, but it was slightly slower than the model in [17].

Surana et al. [25] designed an extended model for the MCPF-tree-based approach [24]. However, the MCPF-tree-based approach did not satisfy the downward closure properties while mining rare full periodic frequent patterns in transactional databases. Hence, the authors proposed another efficient list-based tree data structure, named the maximum constraints periodic frequent pattern tree (MaxCPF-tree), to speed up the mining process. The MaxCPF-tree had a MaxCPF-list and a prefix tree to preserve the transactional identifiers of the patterns. Furthermore, the authors used similar constraint measures to those that were used in the MCPF-tree-based approach, called the minimum item support and maximum item periodicity, to overcome the combinatorial explosion problems that occur while performing the mining. In addition, the authors also used two other pruning techniques while discarding the uninteresting patterns. Therefore, some of the uninteresting patterns that were discovered in the MCPF-tree-based model were pruned from the MaxCPF-tree using these techniques. Finally, the authors showed that the designed approach could extract rare full periodic frequent patterns faster than the MCPF-tree-based model.

Uday et al. [26] designed an interesting novel measure that was named the minimum periodic ratio to discover full periodic frequent patterns in transactional databases. The authors also introduced the concept of potential patterns only consisting of a single item and proposed a novel tree, named extended periodic frequent pattern tree (ExPF-tree), and the so-called extended periodic frequent pattern-growth (ExPF-growth) algorithm to mine the databases. The ExPF-tree had an ExPF-list and a prefix tree to preserve the transactional identifiers of the patterns. In addition, the authors also used two other pruning techniques to discarding the uninteresting patterns.

Uday et al. [27] designed an efficient algorithm, named periodic frequent pattern-growth++ (PFP-growth++), to discover the full periodic frequent patterns in transactional databases. The authors also introduced a tree-based data structure, named periodic frequent pattern tree++ (PF-tree++), to store the patterns and complete the mining process. The PF-tree++ had a PF-list++ and a prefix tree to preserve the transactional identifiers of the patterns. Furthermore, the authors used a novel concept that was named local periodicity to complete the mining process as quickly as possible by using two different phases, which were called the expanding phase and the shrinking phase. Finally, two novel pruning techniques were introduced to complete the mining process efficiently.

Venkatesh et al. [29] designed an extended periodic frequent pattern-growth (EPF-growth) algorithm to discover rare full periodic frequent patterns in non-uniform transactional databases. The authors used an efficient list-based tree data structure, named the extended periodic frequent pattern tree (EPF-tree), to complete the mining process effectively. The EPF-tree had an EPF-list and a prefix tree to preserve the transactional identifiers of the patterns. The authors also used a novel constraint measure, which was named periodic-all-confidence, to extract the interesting rare periodic frequent patterns. Finally, the authors claimed that the designed approach could extract rare full periodic frequent patterns more efficiently compared to different existing models [17,24,25].

In the literature, the model for finding periodic frequent patterns was extended to discover fuzzy periodic frequent patterns [30], local periodic patterns [31], stable periodic patterns [32], top-*k* periodic patterns [18], recurring patterns [33], periodic high-utility sequential patterns [34], non-overlapping sequential pattern mining [35], and periodic sequential patterns [36].

Most of the approaches mentioned above only discover the full periodic frequent patterns using specific constraint measures. Even though the full periodic behavior of a pattern is significant to the user, partial periodic behavior also exists in most real-world scenarios. As a result, it is often helpful to mine partial periodic patterns in temporal databases as well.

Overall, the proposed method [6] for finding partial periodic patterns in temporal databases is novel and distinct from the other existing studies. However, this is a substantially extended version of that method [6].

## 3. Partial Periodic Pattern Model

A complete set of items (e.g., events or symbols) that appeared in a database was represented by $I = \{i_1, i_2, \cdots, i_c\}$, where $c \geq 1$ is the unique items count in the database. A sample set of items $Y \subseteq I$ was called a pattern. We defined a pattern as a $z$-pattern when it contained $z$ items. Furthermore, the length of the pattern was said to be $z$. We let a temporal database $TDB$ be an ordered set of transactions, i.e., $TDB = \{tr_1, tr_2, \cdots, tr_d\}$, where $d = |TDB|$ is the database size ($d$ number of transactions) and $tr_d$, $d \geq 1$ is a transaction in the database. Each transaction $tr_d$ contained three tuples, i.e., the transaction identifier, time stamp, and pattern. Therefore, $tr = (tid, ts, X)$, where $tid$ is the transactional identifier, $ts \in \mathbb{R}$ is the transaction time (or time stamp), and $X$ is the pattern. We let $ts_{min}$ and $ts_{max}$ denote the minimum and maximum time stamps in $TDB$, respectively. Please note that the difference between $(ts_{max} - ts_{min} + 1)$ might not be equal to $|TDB|$ as a temporal database allows time gaps between consecutive transactions and for transactions to share common time stamps. This was contrary to the previous works on finding full periodic frequent patterns in transaction databases [17], as they considered $(ts_{max} - ts_{min} + 1) = |TDB|$. In other words, a temporal database could represent a transactional database but not vice versa. For a transaction $tr = (tid, ts, X)$, such that $Y \subseteq X$, it was said that $Y$ occurred at $tr$ and such a time stamp was denoted as $ts^Y$. We let $TS^Y = (ts_k^Y, ts_l^Y, \cdots, ts_m^Y)$ and $k \leq l \leq m$ be the ordered list of the time stamps of transactions in which $Y$ appeared in $TDB$. The number of transactions that contained $Y$ in $TDB$ (i.e., the size of $TS^Y$) was defined as the *support* of $Y$ and denoted as $sup(Y)$, i.e., $sup(Y) = |TS^Y|$. The complete list of symbols that are used in this paper are shown in Table 1.

**Example 3.** *A temporal database with $I = \{klmnopq\}$ is shown in the Table 2. The pattern km comprised the items k and m. This pattern contained two items; therefore, it was a 2-pattern. The length of this pattern was also two. In the first transaction, $tr_1 = (1001, 1, klm)$, where 1001 is the tid of the transaction, 1 is the time stamp of this transaction, and klm denotes the items that occurred in this transaction. Other transactions in this database were represented in the same way. There were 14 transactions in this database. As a result, $d = 14$. The minimum and maximum time stamps in the database were 1 and 13, respectively. As a result, $ts_{min} = 1$ and $ts_{max} = 13$. The pattern km appeared in the transactions that had time stamps of 1, 3, 4, 6, 9, and 13. As a result, $TS^{km} = \{1, 3, 4, 6, 9, 13\}$. The support of km was $sup(km) = |TS^{km}| = 6$.*

**Table 1.** A complete list of symbols that are used in this paper.

| Notation | Abbreviation |
|---|---|
| $TDB$ | The temporal database that was used in our paper |
| $I$ | Te set of items used in our $TDB$ |
| $c$ | The unique items count in the database |
| $X$ or $Y$ | A pattern consisting of a sub-set of items that was chosen from the complete set $I$ |
| $d$ | The total number of transactions present in a $TDB$ |
| $tid$ | The transactional identifier |
| $ts$ | The transaction time (or time stamp) |
| $TS^Y$ | The ordered list of the time stamps of transactions in which $Y$ appeared in $TDB$ |
| $ts_k^Y$ | The $k$th entry of the pattern $Y$ in the list $TS^Y$ |
| $ts_{min}$ | The minimum time stamp |
| $ts_{max}$ | The maximum time stamp |
| $IAT^Y$ | A list of all inter-arrival times of $Y$ in $TDB$ |
| $iat_k^Y$ | The $k$th entry of the pattern $Y$ in the list $IAT^Y$ |
| $per$ | The user-specified periodicity |
| $\widehat{IAT^Y}$ | The set of all inter-arrival times in $IAT^Y$ that have $iat^Y \leq per(Y)$ |
| $PS(Y)$ | The period support of the pattern $Y$ |
| $minPS$ | The user-specified minimum period support |

**Definition 1 (Periodic appearance of pattern** $Y$**).** *We let $ts_a^Y$, $ts_b^Y \in TS^Y$, and $1 \le a < b \le d$ denote any two consecutive time stamps in $TS^Y$. The time difference between $ts_b^Y$ and $ts_a^X$ was referred to as an inter-arrival time of $Y$ and denoted as $iat^Y$, i.e., $iat^Y = ts_b^Y - ts_a^Y$. We let $IAT^Y = \{iat_1^Y, iat_2^Y, \cdots, iat_b^Y\}$ and $b = sup(Y) - 1$ be the list of all inter-arrival times of $Y$ in TDB. An inter-arrival time of $Y$ was said to be periodic (or interesting) when it was no more than the user-specified period (i.e., per). That is, a $iat_i^Y \in IAT^Y$ was said to be periodic when $iat_i^Y \le per$.*

**Table 2.** The temporal database.

| *tid* | *ts* | *Items* |
|---|---|---|
| 1001 | 1 | $k, l, m$ |
| 1002 | 3 | $o, p, q$ |
| 1003 | 3 | $k, l, m, p$ |
| 1004 | 4 | $k, l, m, p, q$ |
| 1005 | 5 | $k, n, q$ |
| 1006 | 6 | $k, l, m, n, p$ |
| 1007 | 7 | $k, p, q$ |
| 1008 | 7 | $k, l, o, p$ |
| 1009 | 8 | $m, n, q$ |
| 1010 | 9 | $k, l, m, n$ |
| 1011 | 11 | $k, l, p$ |
| 1012 | 12 | $l, p, o$ |
| 1013 | 13 | $k, l, m, n$ |
| 1014 | 13 | $o, p, q$ |

**Example 4.** *The pattern km initially appeared at the time stamps of 1 and 3. The difference between these two time stamps produced an inter-arrival time of km, i.e., $iat_1^{km} = 2 \, (= 3 - 1)$. Similarly, the other inter-arrival times of km were $iat_2^{km} = 1 \, (= 4 - 3)$, $iat_3^{km} = 2 \, (= 6 - 4)$, $iat_4^{km} = 3 \, (= 9 - 6)$, and $iat_5^{km} = 4 \, (= 13 - 9)$. Therefore, the resultant $IAT^{km} = \{2, 1, 2, 3, 4\}$. When the user-specified per $= 2$, then $iat_1^{km}$, $iat_2^{km}$ and $iat_3^{km}$ were considered to be the periodic occurrences of km in the database. The $iat_4^{km}$ and $iat_5^{km}$ were considered to be aperiodic occurrences of km because $iat_4^{km}$ and $iat_5^{km} \not\le per$.*

In the proposed model, we considered an inter-arrival time of $Y$ to be interesting when $iat^Y \le per$. However, our model was adaptable and allowed for other ways of considering the inter-arrival time of a pattern to be interesting. For example, we could consider the inter-arrival time of a pattern to be interesting when $iat^Y \le per \pm \Omega$, where $\Omega > 1$ is a constant that denotes time tolerance. Nevertheless, for brevity, we stuck to the above definition.

**Definition 2 (Period support of pattern** $Y$**).** *We let $\widehat{IAT^Y}$ be the set of all inter-arrival times in $IAT^Y$ that had $iat^Y \le per(Y)$. Therefore, $\widehat{IAT^Y} \subseteq IAT^Y$, such that when $\exists iat_b^Y \in IAT^Y : iat_b^Y \le per(Y)$, then $iat_b^Y \in \widehat{IAT^Y}$. The period support of $Y$ was denoted as $PS(Y) = |\widehat{IAT^Y}|$.*

**Example 5.** *Continuing with the previous example, $\widehat{IAT^{km}} = \{2, 1, 2\}$. Therefore, the period support of km was $PS(km) = |\widehat{km}| = |\{2, 1, 2\}| = 3$.*

As defined previously, the *period support* represented the number of cyclic repetitions of a pattern in the database. In other words, the proposed measure considered both the *support* and *inter-arrival times* in the database to determine the interestingness of a pattern.

**Definition 3** (Partial periodic pattern $Y$). *A pattern $Y$ was a partial periodic pattern when $PS(Y) \ge minPS$, where minPS is the user-specified minimum period support.*

**Example 6.** *Continuing with the previous example, when the user-specified minPS = 2, then km was a partial periodic pattern because PS(km) ≥ minPS.*

**(Problem definition)** Given a temporal database ($TDB$), a set of items ($I$), *period* (*per*), and *minimum period support* (*minPS*), the problem of finding partial periodic patterns involved discovering all patterns in $TDB$ that had a *period support* of no less than *minPS*.

The *support* of a pattern could be expressed as a percentage of $|TDB|$. The *period support* of a pattern could be expressed as a percentage of $|TDB| - 1$, where $|TDB| - 1$ is the maximum period support that a pattern could have in the database. The *inter-arrival times* of a pattern and the *period* could be expressed as a percentage of ($ts_{max} - ts_{min}$). This paper employs the above definitions of *support*, *period support*, *inter-arrival times*, and *period* for brevity.

The partial periodic patterns that were found by the proposed model satisfied the *downward closure properties*. The correctness of our statement is demonstrated by Lemma 1 and is based on Property 1. The following section describes our algorithm, which discovers all of the partial periodic patterns in a temporal database using this property.

**Property 1.** *when $Y \subset X$, then $TS^Y \supseteq TS^X$. Therefore, $PS(Y) \geq PS(X)$.*

**Lemma 1.** *Let X and Y be two patterns, such that $Y \subset X$ and $Y \neq \emptyset$. When X is a partial periodic pattern, then Y is also a partial periodic pattern.*

**Proof.** According to Definition 3, when $X$ is a partial periodic pattern, then $PS(X) \geq minPS$. Based on Property 1, it turns out that $PS(Y) \geq PS(X) \geq minPS$. Henceforth, $Y$ is also a partial periodic pattern. □

## 4. 3P-Growth

Traditional pattern-growth algorithms, which are extensions of FP-growth [8], cannot be used to find partial periodic patterns in an unevenly spaced time series. Therefore, we developed a novel algorithm named 3P-growth to find interesting partial periodic patterns in temporal databases. The proposed algorithm consists of two phases: (*i*) in the initial phase, we scan the entire database and build a tree that is named the partial periodic pattern tree (3P-tree) and (*ii*) in the next phase, we recursively mine the 3P-tree by pruning each one-length partial periodic pattern according to its support to discover the complete set of partial periodic patterns in the temporal database. The 3P-tree structure is explained in the subsequent section.

### 4.1. The 3P-Tree Structure

A 3P-tree consists of two components: a linear list-based data structure, named a 3P-list, and a non-linear tree-based data structure, named a prefix tree. Initially, a 3P-list is constructed by reading the complete database. It consists of two distinct components: (*i*) an *item*, which is denoted as i, and (*ii*) period support, which is denoted as *ps* and also maintains a pointer to store the link of the first node in the prefix tree that carries the item. Even though the overall representation of the items in a 3P-tree looks similar to an FP-tree, i.e., both trees arrange the items according to their support in descending order, the nodes in a 3P-tree are named ordinary nodes and tail nodes. The former is a node that is similar to that used in an FP-tree [8]. In contrast, the latter represents the temporal occurrence information of the last item of any sorted transaction. The 3P-tree maintains a particular data structure, named a *ts*-list, in the tail nodes to preserve the temporal information. Hence, the structure of a *tail* node is $k[ts_a, ts_b, ..., ts_d]$ and $1 \leq a \leq b \leq d$, where $k$ is the item name of the node and $ts_k \in \mathbb{R}$ is the time stamp of the transaction that contains the items from the *root* up to the node $k$. Figure 1 depicts the conceptual structure of a 3P-tree. Each node in a 3P-tree has parent, children, and node traversal pointers, as with an FP-tree. Please note that, unlike an FP-tree, none of the nodes in a 3P-tree preserve the support

count. The items in the prefix tree are organized in descending order of support to permit a high degree of compactness.

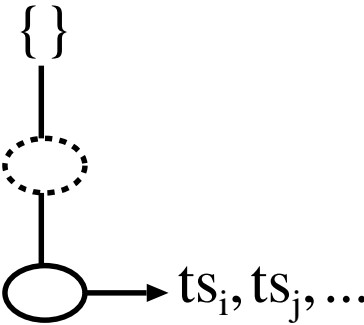

**Figure 1.** Conceptual structure of the prefix tree in a 3P-tree. Dotted ellipses represent ordinary nodes, while the other ellipses represent the tail nodes of the sorted transactions with time stamps of $ts_i$ and $ts_j \in R$.

It can be assumed that the structure of the prefix tree in a 3P-tree may not be memory-efficient since it explicitly preserves the time stamps of each transaction. However, it has been suggested that such a tree could achieve memory efficiency by only retaining transaction information in the tail nodes and omitting the support count field at each node [17]. Furthermore, 3P-trees avoid the complicated combinatorial explosion problem of candidate generation, unlike Apriori-like algorithms [20]. On the other hand, keeping transactional identifier information in a tree can lead to inefficient frequent pattern mining [37] and periodic frequent pattern mining [17].

*4.2. Construction of a 3P-Tree*

The construction of a 3P-tree is a two-step process. First, a 3P-list is built by reading the complete database at once and generating 1-patterns (one-length partial periodic patterns). After that, the prefix tree is built as the generated partial periodic patterns satisfy the anti-monotonic properties. The user-defined parameters *per* and *minPS* are then used to discard the uninteresting (or aperiodic) patterns. Figure 2 demonstrates how Algorithm 1 was used to create a 3P-list for Table 2. In this study, we fixed the values of both *per* and *minPS* at two.

In this study, we used two temporary lists to build the complete 3P-list structure. We let $sup$ be a temporary list variable that was used to hold the *support* information of the items and $ts_l$ also be a temporary list variable that was used to hold the time of the last occurrence of an item, i.e., $k_j \in I$. After reading the first transaction of 1001:1:*klm*, items $k$, $l$, and $m$ were inserted into the 3P-list and their $ps, sup$, and $ts_l$ values were set as $0, 1$, and $1$, respectively (lines 5 and 6 in Algorithm 1). Figure 2a shows the 3P-list that was generated after reading the first transaction. After reading the second transaction of 1002:3:*opq*, items $o$, $p$, and $q$ were inserted into the 3P-list and their $ps, sup$, and $ts_l$ values were set as $0, 1$, and $3$, respectively. Figure 2b shows the 3P-list that was generated after reading the second transaction. After reading the third transaction of 1003:3:*klmp*, the $ps, sup$, and $ts_l$ values of the items $o$, $p$, and $q$ were kept in the 3P-list without any change. In addition, the $ps, sup$, and $ts_l$ values of existing items $k$, $l$, $m$, and $p$ were updated to $1, 2$, and $3$, respectively (lines 8 to 10 in Algorithm 1). Figure 2c shows the 3P-list that was generated after reading the third transaction. After reading the fourth transaction of 1004:4:*klmpq*, the $ps, sup$, and $ts_l$ values of the item $o$ were kept in the 3P-list without any change. In addition, the $ps, sup$, and $ts_l$ values of existing items $k$, $l$, $m$, and $p$ were updated to $2, 3$, and $4$, respectively, and for item $q$, the $ps, sup$, and $ts_l$ values were updated to $1, 2$, and $4$, respectively. Figure 2d shows the 3P-list that was generated after reading the fourth transaction. After reading the fifth transaction of 1005:5:*knq*, item $n$ was inserted into the 3P-list by setting its $ps, sup$, and $ts_l$ values to $0, 1$, and $5$, respectively, and maintaining the $ps, sup$, and $ts_l$ values of the

items $l, m, o$, and $p$ in the 3P-list without any change. In addition, the $ps, sup$, and $ts_l$ values of existing item $k$ were updated to $3, 4$, and $5$, respectively. The $ps, sup$, and $ts_l$ values of existing item $q$ were also updated to $2, 3$, and $5$, respectively. Figure 2e shows the 3P-list that was generated after reading the fifth transaction. After reading the sixth transaction of 1006:6:$klmnp$, the $ps, sup$, and $ts_l$ values of the items $o$ and $p$ were maintained in the 3P-list without any change. In addition, the $ps, sup$, and $ts_l$ values of existing items $l, m$, and $p$ were updated to $3, 4$, and $6$, respectively. The $ps, sup$, and $ts_l$ values of existing item $k$ were also updated to $4, 5$, and $6$, respectively. The $ps, sup$, and $ts_l$ values of existing item $n$ were updated to $1, 2$, and $6$, respectively. Figure 2f shows the 3P-list that was generated after reading the sixth transaction. A similar procedure was followed for the remaining transactions that were available in the database and generated the complete 3P-list structure. The full 3P-list, which was generated after reading the complete database, is shown in Figure 2g. Finally, some of the aperiodic patterns that were available in the 3P-list were pruned based on the user-defined $minPS$ value, i.e., item $o$ was pruned from the 3P-list as its $PS$ value was less than the $minPS$ value. As a result, only one-length partial periodic patterns were displayed and these patterns were sorted in descending order based on their support ($sup$) values (line 11 in Algorithm 1). Figure 2h shows the final 3P-list, which contained a sorted list of all of the partial periodic items. We let $CI$ denote this sorted list of partial periodic items.

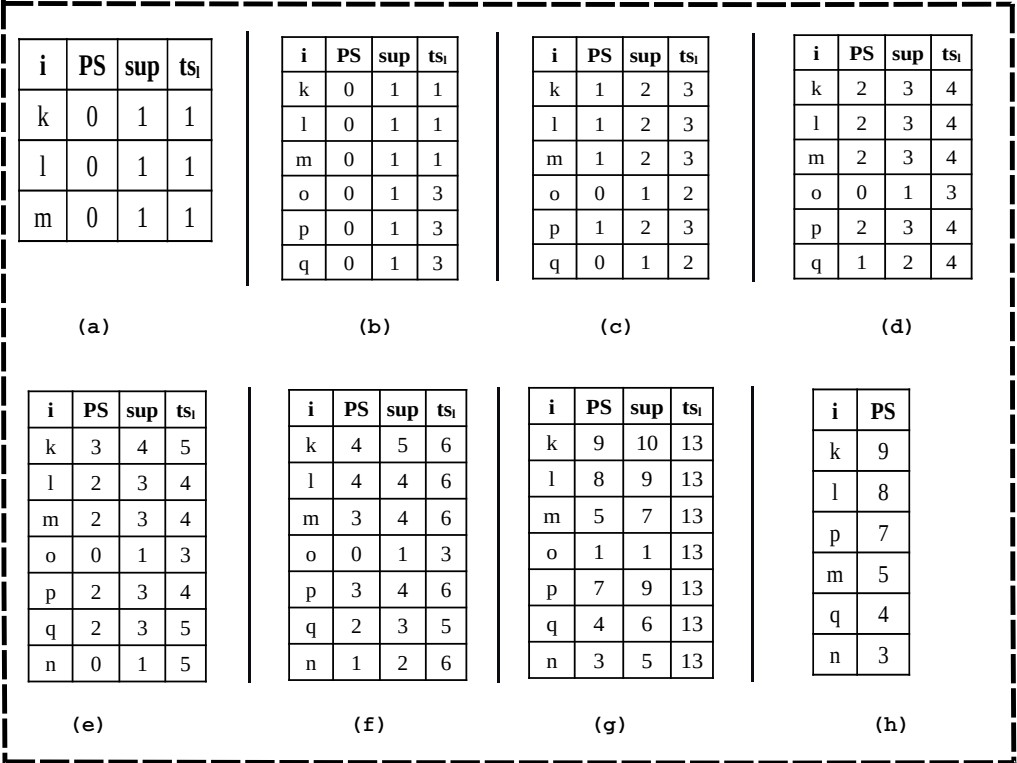

**Figure 2.** Construction of the 3P-List: (**a**) the content of the list after reading the first transaction; (**b**) the content of the list after reading the second transaction; (**c**) the content of the list after reading the third transaction; (**d**) the content of the list after reading the fourth transaction; (**e**) the content of the list after reading the fifth transaction; (**f**) the content of the list after reading the sixth transaction; (**g**) the content of the list after reading the entire database; (**h**) the final 3P-list containing the sorted list of items.

We conducted another scan of the database after discovering the partial periodic items and created the prefix tree of the 3P-tree, as shown in Algorithms 2 and 3. These are the same algorithms that are used to build an FP-tree [8]. The primary distinction is that, unlike an FP-tree, none of the nodes in a 3P-tree keep track of the *support* count.

---

**Algorithm 1** Construction of 3P-list: *TDB*, temporal database; *I*, set of items; *minPS*, minimum period support; *per*, period.

---

1: The *timestamps* of the last occurring transactions of all items in the 3P-list are explicitly recorded in a temporary array called $ts_l$ for each item. Similarly, all items in the 3P-list have their *support* explicitly recorded in another temporary array called *sup*. (To achieve memory efficiency, the 3P-tree will be built in the support descending order of items.) After finding partial periodic items (or 1-patterns), these two arrays can be ignored.

2: Let $t = \{tid, ts_{cur}, Y\}$ denote the current transaction with identifier *tid*, $ts_{cur}$ representing the time stamp, and *Y* as a pattern, respectively;

3: **for** each transaction $t \in TDB$ **do**

4:     **for** each item $j \in Y$ **do**

5:         **if** *j* does not exist in 3P-list **then**

6:             Add *j* to the 3P-list and set $ps(j) = 0$, $sup(j) = 1$ and $ts_l(j) = ts_{cur}$;

7:         **else**

8:             **if** $ts_{cur} - ts_l(j) \leq per$ **then**

9:                 Set $ps(j) + +$;

10:            Set $ts_l = ts_{cur}$ and $sup(j) + +$;

11: Prune any aperiodic items from the 3P-list with a *period support* value of less than *minPS*. Then, consider the remaining items in the 3P-list as partial periodic items, and sort them by *support* in descending order. The symbol *CI* denotes this sorted list of items.

---

The 3P-tree in this study was constructed as follows. We created the root node of the tree and labeled it *null*. Then, we scanned the database once more. The items of each transaction were processed in *CI* order (i.e., sorted according to descending support count). For each transaction, a branch was created so that only the tail nodes recorded the transaction time stamps. For instance, the scan of the first transaction of 1001:1:*klm*, which contained three items (*k*, *l*, and *m* in *CI* order), resulted in the first branch of the tree being built with three nodes: $\langle k \rangle$, $\langle l \rangle$, and $\langle m{:}1 \rangle$, where *m* is linked as a child of the root, *l* is linked as a child of the node *k*, and finally, *m*:1 is linked as a child of the node *l*. Figure 3a shows the 3P-tree that was formed after scanning the first transaction. The second transaction of 1002:3:*opq*, which had items *p* and *q* in *CI* order, resulted in a branch in which *p* was linked as a child of the root and *q*:3 was linked as a child of the node *p*. Figure 3b shows the 3P-tree that was formed after scanning the second transaction. The third transaction of 1003:3:*klmp*, which had items *k*, *l*, *p*, and *m* in *CI* order, resulted in a branch in which *p* was linked as a child of the node *l* and *m*:3 was linked as a child of the node *p*. Figure 3c shows the 3P-tree that was formed after scanning the third transaction. The fourth transaction of 1004:4:*klmpq*, which had items *k*, *l*, *p*, *m*, and *q* in *CI* order, resulted in a branch in which *q*:4 was linked as a child of the node *m*:3. On the other hand, this branch shared the prefix *klpm* with the current path of the third transaction. Figure 3d shows the 3P-tree that was formed after scanning the fourth transaction. The fifth transaction of 1005:5:*knq*, which had items *k*, *q*, and *n* in *CI* order, resulted in a branch in which *n* was linked as a child of the node *k* and *n*:5 was linked as a child of the node *q*. Figure 3e shows the 3P-tree that was formed after scanning the fifth transaction. The sixth transaction of 1006:6:*klmnp*, which had items *k*, *l*, *p*, *m*, and *n* in *CI* order, resulted in a branch in which *n*:6 was linked as a child of the node *m*:3. Figure 3f shows the 3P-tree that was formed after scanning the sixth transaction. The remaining transactions were processed in the same way and the tree was updated accordingly. Figure 3g shows the 3P-tree that was built after scanning the entire database. For clarity, we do not display the node traversal pointers in the trees; however, they were maintained the same way as those in an FP-tree.

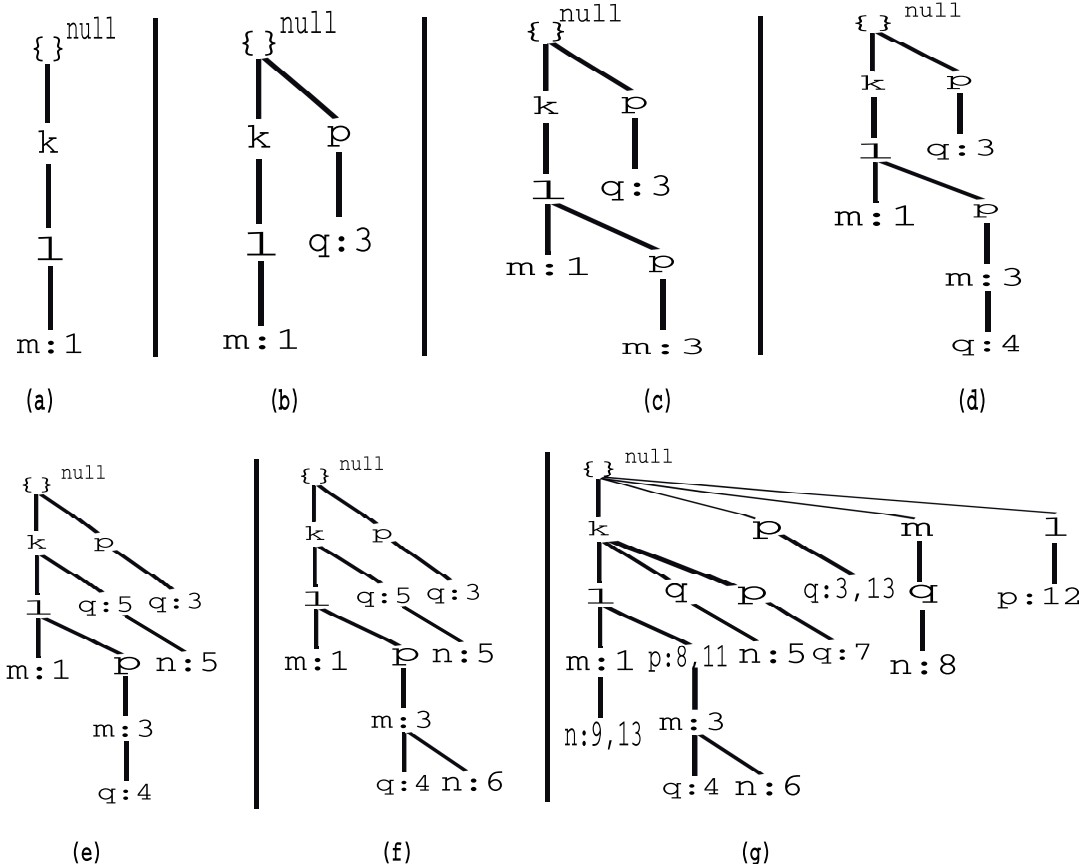

**Figure 3.** Construction of the 3P-tree: (**a**) the tree after reading the first transaction; (**b**) the tree after reading the second transaction; (**c**) the tree after reading the third transaction; (**d**) the tree after reading the fourth transaction; (**e**) the tree after reading the fifth transaction; (**f**) the tree after reading the sixth transaction; (**g**) the final 3P-tree that was generated after reading the entire database.

---

**Algorithm 2** 3P-tree: $TDB$, 3P-list

---

1: A root node is created for the 3P-tree and label it as "*null*";
2: **for** each transaction $t \in TDB$ **do**
3:　　Set the corresponding transaction's time stamp to $ts_{cur}$;
4:　　Choose and sort the partial periodic items in $t$ according to $CI$'s order. Let's say the sorted candidate item list in $t$ is $[p|P]$, with $p$ being the first item and $P$ being the rest of the list;
5:　　Call $insert\_tree([p|P], ts_{cur}, T)$;

---

**Algorithm 3** Insert tree: $[p|P]$, $ts_{cur}$, $T$.

---

1: **while** $P$ is non-empty **do**
2:　**if** $T$ has a child $N$ such that $p.itemName \neq N.itemName$ **then**
3:　　Make a new node $N$. Allow its parent link to point to $T$. Allow its node-link to be linked to nodes that have the same itemName using the node-link structure. $p$ should be removed from $P$;
4: Add $ts_{cur}$ to the leaf node;

---

The 3P-tree stored the complete information of all of the partial periodic patterns in the database. Property 2 was used to determine the correctness, as shown in Lemmas 2 and 3, where $CI(t)$ is the set of all partial periodic items in $t$ for each transaction $t \in TDB$, i.e., $CI(t) = item(t) \cap CI$, and was named as the partial periodic item projection of $t$.

**Property 2.** *For each transaction in a database, a 3P-tree only keeps a complete set of partial periodic item projections once.*

**Lemma 2.** *A 3P-tree can be used to derive a complete set of all of the partial periodic item projections of all transactions in the TDB for a set TDB and user-defined per and minPS values.*

**Proof.** According to Property 1, each transaction $t \in TDB$ is mapped onto just one path in the tree and any path from the *root* up to a *tail* node keeps the complete projection for exactly $d$ transactions, where $d$ is the total number of entries in the *ts*-list of the *tail* node. □

**Lemma 3.** *The size of a 3P-tree (without the root node) in a TDB for the user-specified per and minPS values is bounded by* $\sum_{t \in TDB} |CI(t)|$.

**Proof.** Each transaction $t$ contributes at most one path of size $|CI(t)|$ to the 3P-tree, according to the 3P-tree construction process and Lemma 2. As a result, at best, the overall size contribution of all transactions is $\sum_{t \in TDB} |CI(t)|$. The size of the 3P-tree is significantly smaller than $\sum_{t \in TDB} |CI(t)|$ since there are usually numerous common prefix patterns throughout the transactions. □

*4.3. Mining Partial Periodic Patterns*

Even though the overall representation of items in a 3P-tree is similar to that in an FP-tree, i.e., both the trees arrange the items according to their support in descending order, we could not directly utilize the FP-growth algorithm to mine the 3P-tree because FP-trees do not maintain the temporal information of the transactions. In contrast, 3P-trees maintain a particular data structure, named a *ts*-list, in each tail node to preserve the temporal information. We also designed a novel pattern-growth-based algorithm to generate partial periodic patterns in a bottom-up manner. We utilized the following property and lemma of 3P-trees as part of this algorithm.

**Property 3.** *In a 3P-tree, a tail node keeps track of the temporal occurrence information of the patterns for all nodes in the path (from the tail node to the root), at least in its ts-list.*

**Lemma 4.** *Let $U = \{k_1, k_2, \cdots, k_d\}$ be a path in a 3P-tree where node $k_d$ is the tail node that carries the ts-list of the path. When the ts-list is pushed up to node $k_{d-1}$, then $k_{d-1}$ keeps the temporal occurrence information of the path $U' = \{k_1, k_2, \cdots, k_{d-1}\}$ for the same set of transactions in the ts-list without losing any information.*

**Proof.** According to Property 3, $k_d$ keeps track of the occurrences of path $U'$ in the transactions in its *ts*-list, at least. As a result, the same *ts*-list at node $k_{d-1}$ keeps the same transaction information for $U'$ with no losses. □

In this study, the 3P-tree was mined in the following manner: (*i*) the mining process was initiated with each partial periodic item being named as the initial suffix pattern; (*ii*) subsequently, the conditional pattern base of this pattern was built, i.e., a sub-database that consisted of the sets of prefix paths in the 3P-tree that co-occurred with the suffix patterns was created and its conditional 3P-tree was built to mine recursively; (*iii*) finally, the suffix patterns with the patterns that were generated by the conditional 3P-tree were concatenated, which resulted in the generation of partial periodic patterns.

Algorithms 4 and 5 show the procedure for discovering partial periodic patterns in a 3P-tree. The working of these algorithms was as follows. Starting with the bottom-most item, i.e., $k$, we built the conditional pattern base (or prefix tree) for each partial periodic item in the 3P-list. The prefix sub-paths of node $k$ were accumulated in a tree structure $PT_k$

to construct the prefix tree for *k*. Since *k* was the bottom-most item in the 3P-list, every node in the 3P-tree that was labeled *k* had to be a tail node. Based on Property 3, we explicitly mapped the *ts*-list of each node of *k* onto all of the items in the corresponding path in the temporary array (one for each item) while constructing $PT_k$. The calculation of *period support* for each item in $PT_k$ was made easier using this temporary array (line 2 in Algorithm 4). For example, when item *l* in $PT_k$ had $PS(l) \geq minPS$, we built a conditional tree for it and mined it recursively to find partial periodic patterns (lines 3 to 6 in Algorithm 4 and the entire Algorithm 5). Furthermore, according to Lemma 4, the *ts*-lists were pushed up to the respective parent nodes in the original 3P-tree as well as in $PT_k$ to enable the construction of the prefix tree for the next item in the 3P-list. Following that, all *k* nodes in the original 3P-tree and *k* entries in the 3P-list were deleted (line 7 in Algorithm 4).

---

**Algorithm 4** 3P-growth: *tree*, $\gamma$.

---

1: **for** each $k_j$ in the header of Tree **do**
2:      Generate pattern $\Delta = k_j \cup \gamma$. Collect all of the $k'_j s$ ts-lists into a temporary array, $TS^\Delta$, and calculate $PS(\Delta)$ by calling *calculatePeriodSupport*$(TS^\Delta)$;
3:      **if** $PS\Delta \geq minPS$ **then**
4:          Construct $\Delta$'s conditional pattern base then $\Delta$'s conditional 3P-tree $Tree_\Delta$;
5:          **if** $Tree_\Delta \neq \varnothing$ **then**
6:              call 3P-growth$(Tree_\Delta, \Delta)$;
7:      Prune $k_j$ from the *Tree* and push the $k_j$'s ts-list to its parent nodes;

---

**Algorithm 5** Calculate period support: $TS^\Delta$, a list of time stamps that contained $\Delta$ in the $TDB$.

---

1: Set $PS(\Delta) = 0$;
2: **for** *int* $j = 0; j < TS^\Delta.length - 1; ++j$ **do**
3:      **if** $TS^\Delta[j+1] - TS^\Delta[j] \geq per$ **then**
4:          $++PS(\Delta)$;
5: return $PS(\Delta)$

---

Considering item *n*, which was the last item in the 3P-list, as shown in Figure 2h, we used the 3P-tree shown in Figure 3g to construct a prefix tree for an item *n*, which was called $PT_n$ and is presented in Figure 4a. It was also named as a conditional pattern base and is represented as a tuple entry in the second column of Table 3, under the item *n*. In $PT_n$, there were five items: $k, l, m, p,$ and $q$. Only item *m* fulfilled the condition $PS(m) \geq minPS$. As a result, the conditional tree $CT_n$ from $PT_n$ was built with only one item, *m*, as shown in Figure 4b. It was also named as a conditional 3P-tree and is represented as a tuple entry in the third column of Table 3, under the item *n*. $TS^{mn}$ was generated using the *ts*-list of *m* in $CT_n$. Algorithm 5 was used to calculate the *period support* of *mn*. Since $PS(mn) \geq minPS$, *mn* was treated as a partial periodic pattern and is represented as a tuple entry, along with its *period support*, in the fourth column of Table 3, under the item *n*. Then, as illustrated in Figure 4c, *n* was pruned from the original 3P-tree and its *ts*-lists were moved to its parent. Next, we chose the item *q*, which was the next last item in the 3P-list, as shown in Figure 2h. We used the 3P-tree shown in Figure 4c to construct a prefix tree for an item *q*, which was called $PT_q$ and is presented in Figure 4d. It was also named as a conditional pattern base and is represented as a tuple entry in the second column of Table 3, under the item *q*. In $PT_q$, there were four items: $k, l, m,$ and $p$. Only item *k* fulfilled the condition $PS(k) \geq minPS$. As a result, the conditional tree $CT_q$ from $PT_q$ was built with only one item, *k*, as shown in Figure 4e. It was also named as a conditional 3P-tree and is represented as a tuple entry in the third column of Table 3, under the item *q*. $TS^{kq}$ was generated using the *ts*-list of *k* in $CT_q$. Algorithm 5 was used to calculate the *period support* of *kq*. Since $PS(kq) \geq minPS$, *kq* was treated as a partial periodic pattern and is represented as a tuple entry, along with its *period support*, in the fourth column of Table 3, under the item *q*. Then, as illustrated in Figure 4f, *q* was pruned from the 3P-tree, as shown in Figure 4c, and its *ts*-lists were moved

to its parent. A similar process was repeated until the 3P-list $\neq \varnothing$. The complete mining process of the 3P-tree that is shown in Figure 3h is represented in Table 3.

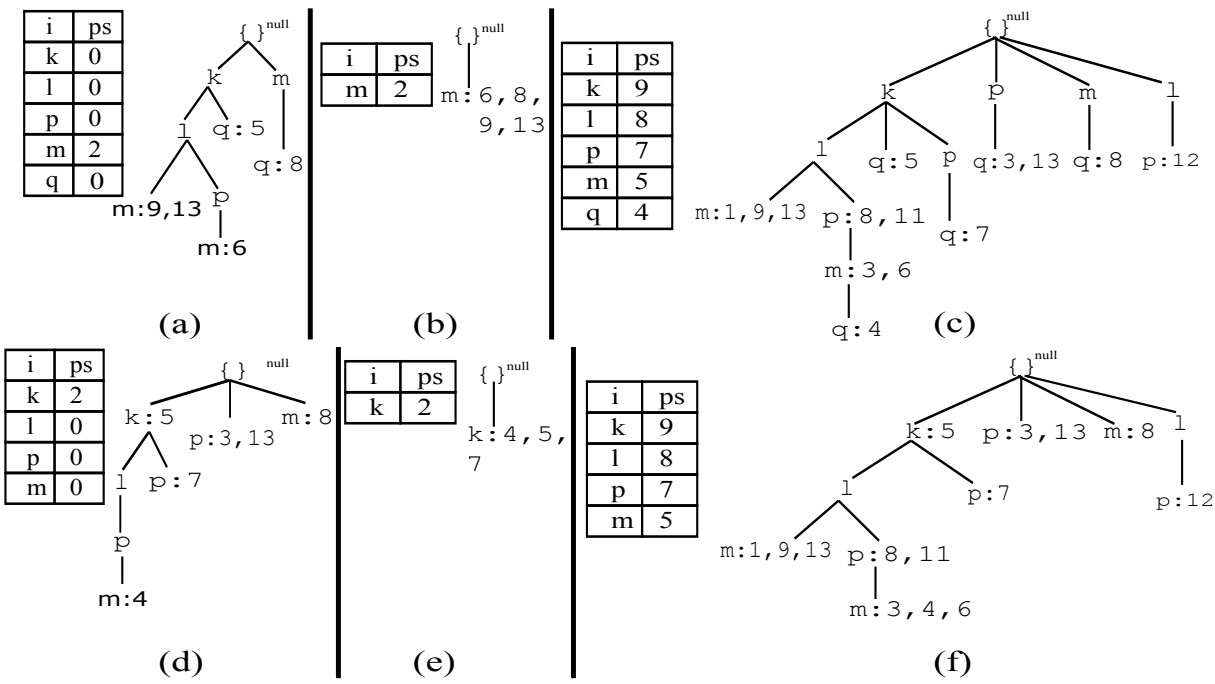

**Figure 4.** Mining the 3P-tree: (**a**) conditional pattern base of item *n*; (**b**) conditional 3P-tree of item *n*; (**c**) 3P-tree after pruning item *n*; (**d**) conditional pattern base of item *q*; (**e**) conditional 3P-tree of item *q*; (**f**) 3P-tree after pruning item *q*.

**Table 3.** The partial periodic patterns that were generated by mining the 3P-tree using the conditional pattern bases.

| *Item* | **Conditional Pattern Base** | **Conditional 3P-Tree** | **Partial Periodic Patterns** |
|---|---|---|---|
| *n* | {*klm*:9, 13}, {*klpm*:6}, {*kq*:5}, {*mq*:8} | $\langle m:6,8,9,13\rangle$ | {*mn*:2} |
| *q* | {*klpm*:4}, {*k*:5}, {*kp*:7}, {*p*:3, 13}, {*m*:8} | $\langle k:4,5,7\rangle$ | {*kq*:2} |
| *m* | {*kl*:1, 9, 13}, {*klp*:3, 4, 6} | $\langle k:1,3,4,6,9,13\rangle$, $\langle l:1,3,4,6,9,13\rangle$, $\langle p:3,4,6\rangle$ | {*km*:3}, {*lm*:3}, {*pm*:2}, {*klm*:3}, {*kpm*:2}, {*lpm*:2}, {*klpm*:2} |
| *p* | {*kl*:3, 4, 6, 8, 11}, {*k*:7}, {*l*:12} | $\langle k:3,4,6,7,8,11\rangle$, $\langle l:3,4,6,8,11,12\rangle$ | {*kp*:4}, {*lp*:4}, {*klp*:3} |
| *l* | {*k*:1, 3, 4, 6, 8, 9, 11, 13} | $\langle k:1,3,4,6,8,9,11,13\rangle$ | {*kl*:7} |

## 5. Experimental Results

In the literature, no algorithm exists that can discover partial periodic patterns in large temporal databases. Therefore, we evaluated the efficiency of the proposed algorithm by varying the *per* and *minPS* parameters and also showed that the proposed algorithm is highly scalable.

### 5.1. Experimental Setup

The 3P-growth algorithm was written in Python and executed on an Intel I5 2.66 GHz machine with 8 GB of memory. The operating system of our machine was Ubuntu 18.04. The 3P-growth algorithm was evaluated using both synthetic (T10I4D100K and T10I4D1000K) and real-world (FAA-incidents, Pollution, and Congestion) databases.

The T10I4D100K and T10I4D1000K databases are sparse synthetic databases that were produced by the IBM data generator [20]. Pattern mining techniques are frequently

evaluated using this data generator. The *tid* of a transaction also represents its time stamp in these databases. The T10I4D1000K is a massive sparse database. We utilized this database to test the scalability of the 3P-growth algorithm.

The FAA incidents database comprises aircraft incidents that were reported to the Federal Aviation Authority (FAA) between 1 January 1978 and 31 December 2014. The FAA raw data included both numerical and category information. We only considered the categorical attributes for our experiments, namely aircraft series, aircraft engine make, aircraft model, aircraft make, primary flight type, operator, event city, event type, aircraft damage, flight conduct code, flight phase, event airport, flight plan filed code, local event date, and PIC certificate type. Any missing values for these attributes were ignored while creating this database.

Air pollution is the leading cause of many of the cardio-respiratory problems that are reported by Japanese residents. For this reason, the Japanese Ministry of Environment established the Atmospheric Environmental Regional Observation System (AEROS) [38] to combat pollution. This system consists of multiple air pollution measurement sensors, which are spread across Japan. Each station collects data about various air pollutants, i.e., $PM_{2.5}$, $NO_2$, and $O_3$ , on an hourly basis. In this experiment, we confined air pollution to $PM_{2.5}$ to show the pollution levels at each location. We collected the raw data hourly from each sensor and converted it into a temporal database, which was named **Pollution**. It is a high-dimensional real-world dense database, which contains 1600 items and 720 transactions.

Monitoring traffic congestion in smart cities is a difficult but critical problem for intelligent transportation systems. For this reason, the Japan Road Traffic Information Center (JARTIC) installed several sensor networks to monitor congestion in several smart cities. The data that are generated by this sensor network represent a spatio-temporal database. For our experiment, we employed the traffic congestion data that were generated by the sensor network that is located in Kobe, Japan. Each transaction in this database was a 5-min interval and contained the following information: *time stamp at 5-min intervals and road segment identifiers that reported congestion of more than 300 m*. The data were collected from 1 July 2015 to 31 July 2015. The **Congestion** database contains 1414 items and 8928 transactions.

The complete statistics of the databases that were used in our experiments are shown in Table 4. It can be observed that the Pollution and Congestion databases are high-dimensional databases that contain long transactions.

**Table 4.** The complete statistics of the databases that were used in our experiments: the minimum, average, and maximum transactions of the databases are represented by min., avg., and max., respectively.

| S. No | Database | Type | Nature | Transaction Length (in Count) | | | Database Size (in Count) |
|-------|----------|------|--------|------|------|------|--------------------------|
| | | | | Min. | Avg. | Max. | |
| 1 | T10I4D100K | Synthetic | Sparse | 1 | 10 | 29 | 1,00,000 |
| 2 | T10I4D1000K | Synthetic | Sparse | 1 | 10 | 31 | 9,83,155 |
| 3 | FAA Incidents | Real | Sparse | 2 | 12 | 14 | 78,864 |
| 4 | Congestion | Real | Sparse | 1 | 58 | 337 | 8928 |
| 5 | Pollution | Real | Dense | 11 | 460 | 971 | 720 |

### 5.2. Evaluation of 3P-Growth

Figure 5a–d show the number of partial periodic patterns that were generated using 3P-growth in the T10I4D100K, Congestion, Pollution, and FAA incidents databases for various *per* and *minPS* values. These figures allowed us to make the subsequent observations: (*i*) when we increased the value of *per*, then the total number of partial periodic patterns that were generated in each of the databases could also increase because when we increased the *per* constraint, most of the aperiodic patterns became partial periodic patterns; (*ii*) an increase in *minPS* could result in a decrease in the number of partial periodic patterns

because increasing the *minPS* value increased the minimum number of cyclic repetitions that were required for a pattern to be considered as a partial periodic pattern.

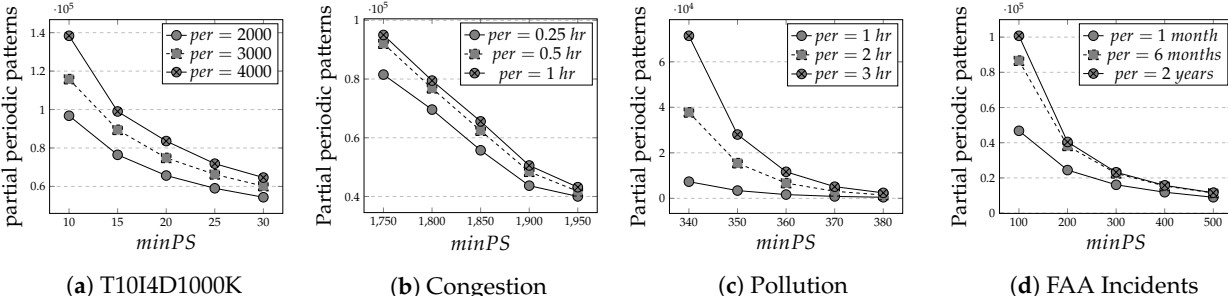

**Figure 5.** Number of partial periodic patterns that were generated in the various databases (**a**–**d**) for different *minPS* and *per* values.

Figure 6a–d show the runtime requirements of the 3P-growth algorithm in the T10I4D100K, Congestion, Pollution, and FAA incidents databases for various *per* and *minPS* values. These figures allowed us to make the subsequent observations: (*i*) the runtime requirements of 3P-growth could increase as the *per* value increased, which was because 3P-growth had to find more partial periodic patterns with the increase in *per*; (*ii*) an increase in *minPS* could decrease the runtime requirements of the 3P-growth algorithm because as the *minPS* value increased, 3P-growth had to find fewer partial periodic patterns.

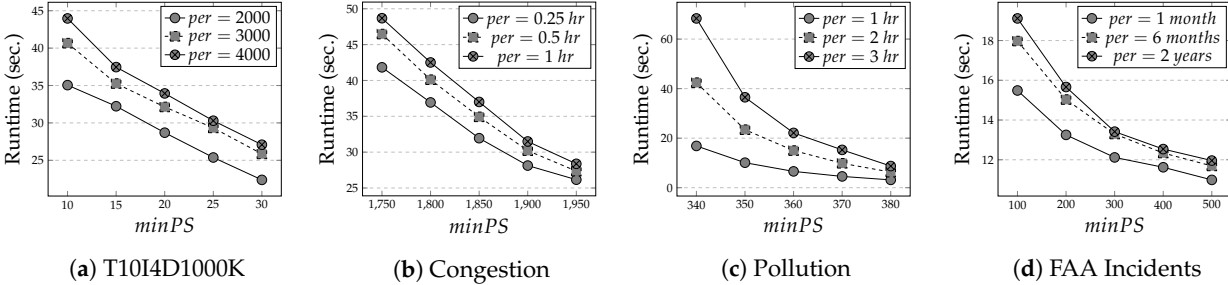

**Figure 6.** Runtime requirements of the 3P-growth algorithm in the various databases (**a**–**d**) for different *minPS* and *per* values.

Figure 7a–d show the memory consumption details of the 3P-growth algorithm in the T10I4D100K, Congestion, Pollution, and FAA incidents databases for various *per* and *minPS* values. These figures allowed us to make the subsequent observations: (*i*) the memory requirements of 3P-growth could increase as the *per* value increased because as the *per* value increased, 3P-growth had to find a greater number of partial periodic patterns; (*ii*) an increase in *minPS* could reduce the memory requirements of the 3P-growth algorithm because as the *minPS* value increased, 3P-growth had to find fewer partial periodic patterns. Changes in the *per* and *minPS* values had the same runtime effects as the generation of partial periodic patterns.

Overall, it could be observed from the above three results that the 3P-growth algorithm could find a vast number of partial periodic patterns in massive databases, even at low *minPS* values and high *per* values.

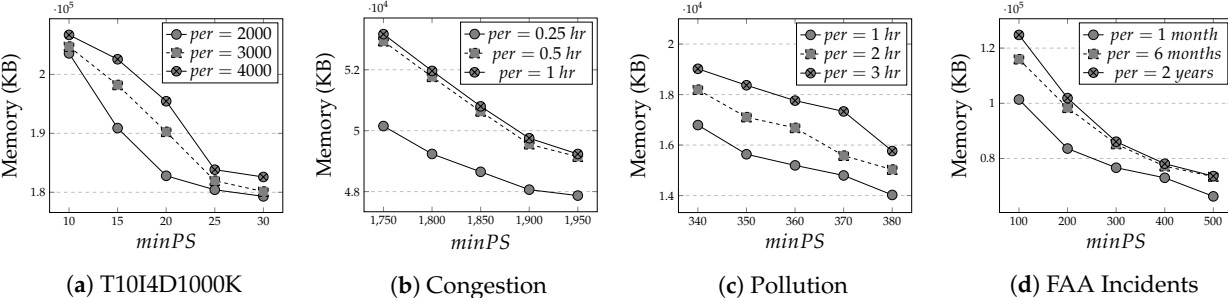

**Figure 7.** Memory consumption details of the 3P-growth algorithm in the various databases (**a**–**d**) for different *minPS* and *per* values.

### *5.3. Scalability of 3P-Growth*

We also investigated the scalability of the 3P-growth algorithm in terms of runtime and memory by varying the length of the temporal database. We used a large temporal database named T10I4D1000K to carry out the scalability task. Initially, it was divided into five equal portions, with each portion consisting of 0.2 million transactions. Next, we investigated the performance of the 3P-growth algorithm at each iteration. We accumulated the previous portion of transactions to the present iteration in order to effectively generate the partial periodic patterns. The values for *per* and *minPS* were chosen as 1500 and 0.05%, respectively. The runtime and memory requirements of 3P-growth for the different sizes of the T10I4D1000K database are shown in Figure 8a,b, respectively. These graphs showed that the overall tree construction, mining time, and memory requirements increased as the database grew. However, the 3P-tree exhibited a stable linear increase in runtime and memory consumption in terms of database size. As a result of the scalability test, it was found that the 3P-growth algorithm could mine partial periodic patterns across massive databases and different items with significant runtime and memory requirements.

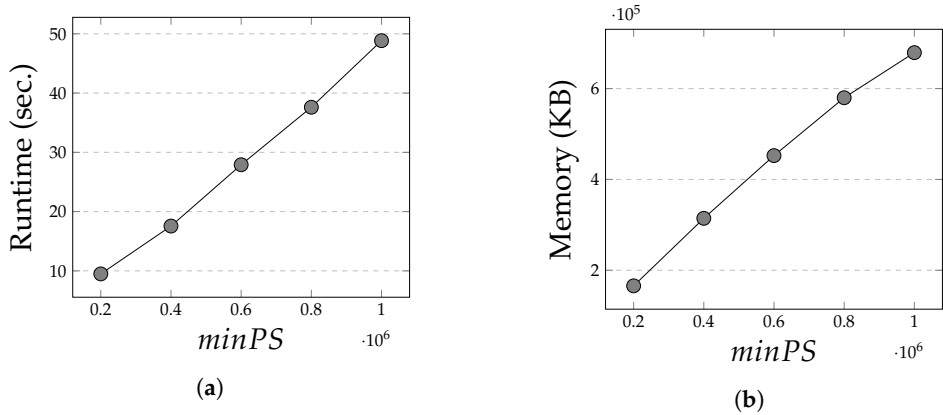

**Figure 8.** Scalability of the 3P-growth algorithm: (**a**) Runtime and (**b**) Memory.

### *5.4. Case Study 1: Interesting Patterns that were Generated in Pollution Database*

Japan's Ministry of Environment set up the Atmospheric Environmental Regional Observation System to combat air pollution. The spatial locations of the sensors (or air pollution monitoring stations) that are situated throughout Japan are shown in Figure 9a. The raw data that are produced from these sensors for a particular air pollutant, i.e., $PM_{2.5}$, are shown in Figure 9b. These raw data were transformed into a temporal database by grouping together the sensor identifiers whose $PM_{2.5}$ values were greater than 16 μg/m$^3$. The generated temporal database is shown in Figure 9c. The proposed 3P-growth algorithm took this temporal database as input and discovered all of the partial periodic patterns within it (see Figure 9d). Figure 9e shows some of the partial periodic patterns that were generated in the temporal database. The spatial visualization of these patterns is shown in Figure 9f. It was observed that people living in southern Japan are regularly exposed to high levels of $PM_{2.5}$. This information could benefit an environmentalist for various

purposes, including the development of policies to control industrial emissions and alert systems for (elderly) people who are sensitive to air pollution.

**Figure 9.** Air pollution data analytics (**a**–**e**). The term *SID* stands for sensor identifier.

*5.5. Case Study 2: Interesting Patterns that were Generated in the Congestion Database*

The spatial locations of the sensors (or road segments) in each of these patterns are shown in Figure 10a for the networks that were established in Kobe, Japan. The raw traffic congestion data that were collected from these sensors are shown in Figure 10b. These raw data were converted into a temporal database by grouping together the road segments whose congestion was more than 300 m. The proposed 3P-growth algorithm (see Figure 10d) accepted this temporal database as input. It output a set of highly congested road segments on which people regularly experienced traffic congestion. Some of the interesting patterns that were generated in the temporal database are shown in Figure 10e. The spatial locations of these patterns are shown in Figure 10f. The generated data could benefit users for various purposes, including urban planning and traffic monitoring (especially during disasters).

Several statistical and machine learning models [39,40] have been developed to predict on-road congestion segments. The proposed model could be employed on the data that are generated by those prediction models to discover sets of highly congested road segments. As a result, for people who regularly experience traffic congestion, such knowledge on heavily congested roads could help traffic control rooms to divert traffic, suggest police patrols, and alert pedestrians on the roads.

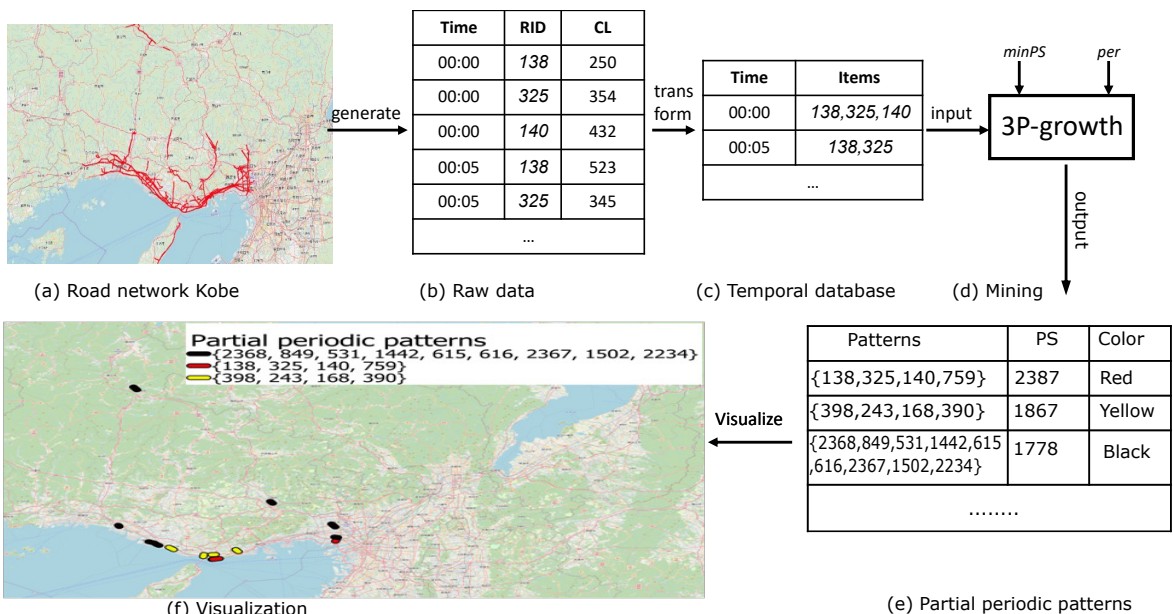

(a) Road network Kobe      (b) Raw data      (c) Temporal database      (d) Mining

(f) Visualization      (e) Partial periodic patterns

**Figure 10.** Congestion data analytics (**a**–**e**). The terms *RID* and *CL* stand for road identifier and congestion length, respectively.

## 6. Conclusions and Future Work

In this paper, we proposed a novel model for finding partial periodic patterns in temporal databases. Two constraints (minimum period support and periodicity) were utilized to find all of the desired patterns. An efficient pattern-growth algorithm was also proposed in this paper, which could enumerate all of the desired patterns efficiently. We experimented with synthetic and real-world databases and from these experiments, we could conclude that the proposed algorithm is memory- and runtime-efficient and highly scalable. Finally, we presented two case studies to demonstrate the effectiveness of the proposed patterns in real-world applications: one on air pollution analytics and one on traffic congestion analytics.

There are several opportunities for future work. Firstly, our investigation was limited to the extraction of periodic patterns in static temporal databases. On the other hand, the method that was proposed here could be applied to the incremental mining of temporal databases. Secondly, the data that are produced from several real-world applications exist naturally as data streams in the era of big data. As a result, it is worth looking into the problem of finding partial periodic patterns in data streams. Thirdly, it is worthwhile exploring alternative measures of period support to meet user and/or application requirements.

**Author Contributions:** Conceptualization, R.U.K. and P.V.; Data curation, R.U.K. and P.V.; Formal analysis, R.U.K. and P.V.; Funding acquisition, R.U.K. and P.V.; Investigation, R.U.K. and P.V.; Methodology, R.U.K. and P.V.; Project administration, R.U.K. and P.V.; Resources, R.U.K. and P.V.; Software, R.U.K. and P.V.; Supervision, R.U.K., P.V., K.Z., H.S., M.T., M.K. and P.K.R.; Validation, R.U.K., P.V., P.R. and C.S.; Visualization, R.U.K. and P.V.; Writing—original draft, R.U.K., P.V. and P.R.; Writing—review & editing, R.U.K., P.V. and P.R. All authors have read and agreed to the published version of the manuscript.

**Funding:** This research was funded by JSPS Kakenhi 21K12034.

**Conflicts of Interest:** The authors declare no conflict of interest.

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
