# Peer review of "Efficient Discovery of Partial Periodic Patterns in Large Temporal Databases"

_electronics, doi:10.3390/electronics11101523_

Round 1
Reviewer 1 Report
This paper proposes an algorithm named Partial Periodic Pattern-growth (3P-growth) for mining partial periodic patterns in the temporal database by using two measures: minPS and per. The experiment was conducted on both synthesis and real-life datasets to evaluate the performance of the proposed algorithm in terms of runtime, memory usage, and scalability.
The proposed algorithm was also applied in two case studies which are air pollution analytics and traffic congestion analytics.
In general, the paper is well-written and structured. The topic conducted in the paper is also interesting. I like the way that the authors show examples of definitions used in the paper. The results seem to be reasonable. Moreover, the proposed algorithm is applied to a real-life application. It finally can detect some interesting and meaningful patterns. Thus, I think the paper matches the level of acceptance. In support of this paper, the authors should make a minor revision to further improve the quality of the paper
- In section 3, insert a table of notations, and abbreviations used in the paper.
- In the experiment, I suggest the authors evaluate the performance of 3P-growth with different sizes of experimental datasets.
Author Response
- In section 3, insert a table of notations, and abbreviations used in the paper.
Reply: Respecting the reviewer suggestion, we have inserted a table of notations and abbreviations in section 3 paragraph 1
- In the experiment, I suggest the authors evaluate the performance of 3P-growth with different sizes of experimental datasets.
Reply: Thank you for your comment, we have conducted experimentations on different database sizes only and the complete statistics of the databases used in our experimentation were shown in the modified version of table 4 in the section 5.
Reviewer 2 Report
This paper focuses on mining partial periodic patterns in a temporal database and proposes an efficient algorithm named 3P-growth. The experimental results verify the performance of the proposed algorithm. The topic is interesting, and the paper is well-structured. However, the drawbacks cannot be neglected.
- In my opinion, the Abstract should be improved, since the background is too long and the principle of the proposed algorithm is too short.
- The toy example in the Introduction section is difficult to understand. More importantly, the toy example should be that periodic pattern mining cannot solve this problem well, while partial periodic pattern mining can.
- The background introduced is too narrow, and the authors pay less attention to the related progress of pattern mining, since the literature published in the last two years have not been cited. For example, the following papers are related to pattern mining methods.
[1] Youxi Wu, Meng Geng, Yan Li, Lei Guo, Zhao Li, Philippe Fournier-Viger, Xingquan Zhu, Xindong Wu. HANP-Miner: High average utility nonoverlapping sequential pattern mining. Knowledge-Based Systems. 2021. 229, 107361
The method in [1] focused on high average utility nonoverlapping sequential pattern mining with periodic gap constraints, which can be used in purchase recommendations. Moreover, the methods focusing on weak-gap strong pattern mining with self-adaptive gap and three-way sequential pattern mining with periodic gap constraints can be used in time series analysis and are a new research direction, worthy of attention.
- The contributions of this paper should be improved, since it would be better to clarify the purpose of this paper.
- Finally, section 6--->Finally, Section 6.
- because iat^{km}_4 andiat^{km}_5--->because iat^{km}_4 and iat^{km}_5
- In Property 1, \subset is undefined symbol.
- The example in Pages 9-10 is too long. The author should select several key points to illustrate the example.
- The figures should be improved. For example, text and picture overlap in Figs. 5, 6, 7, and 8.
- Some figures are blurred. More importantly, Fig. 10 (f) is distorted.
Based on the above drawbacks, this paper should be a major revision.
Author Response
- In my opinion, the Abstract should be improved, since the background is too long and the principle of the proposed algorithm is too short.
Reply: Respecting the reviewer suggestion, we have updated the content of the abstract
- The toy example in the Introduction section is difficult to understand. More importantly, the toy example should be that periodic pattern mining cannot solve this problem well, while partial periodic pattern mining can.
Reply: Respecting the reviewer suggestion, we have removed the example in the introduction section and the same was explained with a real world case study 1 in the section 5.4 along with the visualization of the partial periodic patterns in Figure 9.
- The background introduced is too narrow, and the authors pay less attention to the related progress of pattern mining, since the literature published in the last two years have not been cited. For example, the following papers are related to pattern mining methods.
Reply: Thank you for your valuable suggestion, we have extended the related work with necessary citations.
[1] Youxi Wu, Meng Geng, Yan Li, Lei Guo, Zhao Li, Philippe Fournier-Viger, Xingquan Zhu, Xindong Wu. HANP-Miner: High average utility nonoverlapping sequential pattern mining. Knowledge-Based Systems. 2021. 229, 107361
The method in [1] focused on high average utility nonoverlapping sequential pattern mining with periodic gap constraints, which can be used in purchase recommendations. Moreover, the methods focusing on weak-gap strong pattern mining with self-adaptive gap and three-way sequential pattern mining with periodic gap constraints can be used in time series analysis and are a new research direction, worthy of attention.
- The contributions of this paper should be improved, since it would be better to clarify the purpose of this paper.
Reply: Thank you for the suggestion, we corrected the same
- Finally, section 6--->Finally, Section 6.
Reply: Thank you for the suggestion, we corrected the same.
- because iat^{km}_4 andiat^{km}_5--->because iat^{km}_4 and iat^{km}_5
Reply: Thank you for the suggestion, we corrected the same
- In Property 1, \subset is undefined symbol.
Reply: We have defined the all the necessary symbol in section 3 first paragraph and a new notation table is also introduced inside the paper.
- The example in Pages 9-10 is too long. The author should select several key points to illustrate the example.
Reply: Thank you for the suggestion; taking into the account of the naïve user point of view, we have discussed the example with detailed step by step explanation and all the steps of most crucial ones. Therefore, we are unable to remove any one of those.
- The figures should be improved. For example, text and picture overlap in Figs. 5, 6, 7, and 8
Reply: Thank you for the suggestion, we corrected the same
- Some figures are blurred. More importantly, Fig. 10 (f) is distorted.
Reply: Thank you for the suggestion, we corrected the same
Based on the above drawbacks, this paper should be a major revision.

Round 2
Reviewer 2 Report
The authors carefully revised the manuscript. In my opinion, this paper can be accepted.